# Revolutionizing Molecular cloning: Introducing FastCloneAssist, a Streamlined Python tool for optimizing primer design in restriction & ligation-independent PCR cloning

**Pradip Kumar Singh**(iD)**, Michael S. Donnenberg**(iD)*

Virginia Commonwealth University, Richmond, Virginia, United States of America

* michael.donnenberg@vcuhealth.org

## Abstract

FastCloning, a paradigm shift in PCR cloning, has streamlined the process by eliminating laborious, multi-step traditional methods. This innovative technique, pioneered by Li et al. (2011), utilizes overlapping PCR primers and DpnI digestion for seamless integration of insert DNA into any desired vector position, regardless of restriction sites. This versatility makes FastCloning ideal for constructing fusion proteins, chimeric cDNAs, and manipulating genes with unparalleled ease.

However, efficient primer design remains a critical hurdle, particularly for newcomers, as errors can lead to failed cloning attempts. To address this bottleneck, we present Fast-CloneAssist, a user-friendly Python program that automates FastCloning primer design with minimal user input. Users simply provide vector and insert sequences, along with the desired melting temperature (Tm), and FastCloneAssist provides best primer pairs after calculating optimal primer parameters for efficient PCR amplification and seamless DNA integration using established bioinformatics libraries. This open-source, freely available tool simplifies and accelerates cloning, making this powerful technique accessible to researchers of all levels and expediting scientific discovery.

## Introduction

Molecular cloning, a fundamental pillar of diverse biological research, often involves laborious multi-step procedures demanding significant time and resources [1]. Multiple methods have been developed around the basic concept of restriction digestion and ligation to reduce the labor and improve the efficiency with some success [2–5]. In an alternate approach, FastCloning emerged as a transformative PCR-based location-specific cloning method, eliminating the need for restriction enzymes and ligation while facilitating rapid fragment integration [6]. Briefly, to create a standard vector plus insert chimera, this method uses custom-designed primers to amplify both the vector and insert DNA. Alternatively, for deletions or small

**Data availability statement:** All relevant data are within the manuscript and its Supporting Information files.

**Funding:** The research used funds provided by Virginia Commonwealth University School of Medicine. Accelerate Grant.

**Competing interests:** NO authors have competing interests.

fragment insertions (<120bp) at a desired location, only one set of primers is necessary. After amplification, both scenarios require DpnI treatment to selectively digest parental templates, aiding in in vivo ligation by a mechanism that remains under investigation. Please see the cited paper, Li et al. (2011), for more details.

While FastCloning offers undeniable advantages, meticulous primer design remains crucial for successful amplification and seamless integration of the target fragment. Traditional methods often entail painstaking manual calculations and sequence adjustments to achieve the desired primer Tm, length, and overlap sequences. The process can be time-consuming, with unsuccessful attempts leading to delays. A single base error has the potential to unravel all efforts, highlighting the critical nature of precision and accuracy in this endeavor. Although there are tools available to aid in the design of PCR and restriction cloning specific primers, to date none is available to design experiment-specific FastCloning primers. The online primer tools lack customization options and often require coding expertise, limiting their accessibility to novice users.

To address these challenges, we introduce FastCloneAssist (patent pending), a user-friendly Python script designed to streamline FastCloning primer design. Requiring minimal user input (vector and insert sequences), FastCloneAssist automates the selection of optimal primer lengths, melting temperatures, and overlap sequences, ensuring efficient amplification and in vivo ligation.

### Key features of FastCloneAssist include

**Automated optimization.** Optimizes primer lengths, melting temperatures, and overlap sequences based on user-provided sequences, eliminating tedious manual calculations.

**Customizable Tm.** Offers both default and user-defined Tm options, allowing for flexibility in primer design.

**Ease of use.** Requires little-to-no coding knowledge, making it accessible to users of all skill levels.

**Platform independent.** Runs easily on a computer's local Python environment as well as in cloud computing using Google Colab [7].

By streamlining primer design and overcoming technical barriers, FastCloneAssist empowers researchers to leverage the full potential of FastCloning, potentially accelerating diverse molecular biology applications.

## Method and applications

The primer design scheme is shown in Fig 1. The FastCloneAssist divides FastCloning primers into two broad classes. **Class 1**: this generates primer pairs for a chimera creation where an insert DNA (> 100bp) will be PCR amplified and inserted into a PCR linearized vector DNA (Fig 1A and 1B). Therefore, it generates a total of two pairs of primers, one pair for insert DNA and one for vector DNA. **Class 2**: this generates a single primer pair which can be used to delete a region (Fig 1C) or insert or delete and insert (Fig 1D). Note, the insert for class 2 should be less than 120 bp.

The tool uses the following dependencies and libraries:

○ Biopython: Provides tools for sequence manipulation and translation [8,9].

○ Biopython> SeqUtils.MeltingTemp: Specifically used for Tm calculations [9].

○ primer3: Used for primer design and Tm calculations [10].

○ Bio.SeqUtils.MeltingTemp: Specifically used for Tm calculations [9].

○ Tkinter: Used to create the optional GUI interface [11].

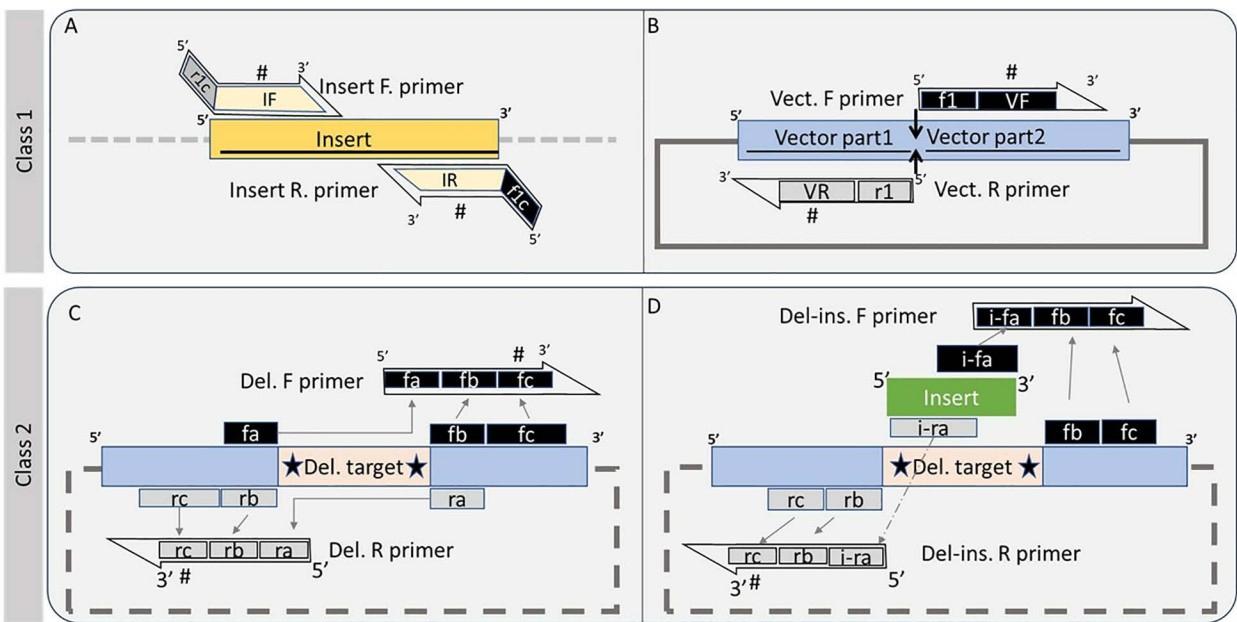

**Fig 1. FastCloneAssist primer design scheme.** A and B, schematics for class 1 primer designing. (A) insert and (B) vector region primer design schemes. Black and gray boxes show selected bases for the forward and reverse primers while the vertical arrows indicate insertion points. Note, box f1c and r1c (A) are exact reverse complements of box f1 and r1 (B), respectively. The size of all other boxes, marked #, is adjusted according to Tm values. C and D, Class 2 primer designing scheme. (C) for deletion primer design, sequences from region box fa & fb should be complementary to box rb and ra respectively and total length of each complementary boxes should be ~8 bp. (D), showing deletion & insertion primer design. The sequence from box i-fa and i-ra should have 16 base pair complementary sequence. Gray arrow marking region of primers sequence and "#" symbol indicates where FastCloneAssist tool adjust length to adjust Tm values.

**Inputs and sequence processing. User input.** The user provides the following inputs through the script's interface:

- Sequences: see the input sequence format below is the class description

- A choice to customize the desired Tm range for primer design (default range is 55 – 65 °C).

**Sequence processing**

- The script splits the input sequence into specific parts as marked by the symbols (see class descriptions)

- It removes any gaps or line breaks from the sequences.

- It generates protein translation into three frames (+1, +2, +3) for the final constructed DNA sequence.

**Primer Design**

- The script calculates the initial Tm values for potential primer sequences based on their nucleotide content.

- It adjusts primer lengths iteratively to achieve Tm values within the desired range (either the default or user-specified range).

- The script determines the appropriate locations for primers based on the boundaries of the vector parts and insert.

**Output.** FastCloneAssist generates and displays or asks to save the following output.

○ Designed primer sequences, along with their Tm values and lengths.

○ FASTA format of the final construct sequence

○ Translations in all three + frames

○ Input information for the record keeping.

Note: Script will ask to save the report as.txt file when used in a local Python environment and shows results and saves in a downloaded folder in the Google Colab environment.

## Workflow detail

**Primer designing for Class 1: Vector and insert specific primers.** Here, the user is asked to provide DNA sequences in the following format:

Vector part-1 + Insert + Vector part-2.

In a word file generate your desired construct, here, place the insert sequence into the vector sequence at the desired location and mark it by "+" symbol, see example below. Note the vector part-1 and vector part-2 should be no less than 40 bp and insert should be 100 bp or more.

Example input sequence format:

TTGGCTGAAGCGATGAGGGGGTGGGTACCTTTTGACGAACTGTCAATAATTT CCGCCGGGGAAATCTCAGGTAATGTTCACCAGGCGTTGGATGATATCATTTA TATGAATGATACAAAAAAGAAAGTAAAA +gcgcactggcagggattatttatcctgtagtcctgcttc tgacgacatgtctgtatttgcatatatttggaactcaggttgttccggcattttcaggcatcctgcctgtagagaaatggcagggcgcag- gcaggactatgtattatcttgctgtattcgttcaggattatcttgtcattacactgctgtctttatgatggtgatattattaatactggca+ ACTCTGTCAAGATGGACCGGAAGGTTGCGTCTCTTCTTTGACAGATTTATTCCCT- GGTCGATATACAAAACCATTATAGGATGTGGTTTCTTGTTATCACTGGCATCGCT- TATTAATGCAGGTATCCCTGTACCGG.

The insert sequence is in lower case (for visuals only), joined by "+" symbols into the vector part-1 and vector part-2 at 5' and 3' of the insert sequence respectively.

To design primers for the vector amplification, "vector part-1" and "vector part-2" sequences are used. "Vector part-1" region specific primer was designed by selecting nucleotides from 3' ends toward 5' of the sequence and the primer length adjusted to meet the desired Tm using SeqUtils modules from biopython. The reverse complement of this sequence is used as the reverse primer for the vector. The forward primer for the vector comes from the "vector part-2" nucleotides, which are selected from the 5' region of the sequence to correspond with the Tm and length of primer. Primers for the insert sequence come from the 5' and 3' terminals of the insert sequence and produce a 16 bp complementary overhang from vector regions. The forward primer of the insert (IF) overhangs (r1c) the vector part-1, whereas the reverse primer of the insert (IR) overhangs (f1c) with the vector part-2, see Fig 1. The length of IF and IR (without their overhangs) is also adjusted to meet the desired Tm values. See the FC_class1 report in S1 File.

**Primer designing for Class 2: Deletion or small fragment insertion.** For class 2, the script designs primers for deletion, insertion and/or deletion & insertion.

Input sequence format for "deletion and insertion" primer designing: Here, user asked to provide DNA sequences in a format where the deletion region (start & end) will be marked by * symbols and the insert sequence is provided at the end marked by * symbol. See below example.

AGCAGCGGCGAGAATCTTTATTTTCAGGGCCATATGAAAGAAAAATTA AATCGTTTATTATTTACTTCTAAAACTCGTATGCGTGTTTTTTTCTAAATTATCTC

GTTATTTATCTAATGGTGTTCCTGTTACTTTTGCTTTAGCTGAATTATATA-
AATTTACTTCTGATGAAGGTCGTAAAAAAGATAATCCTGATGCTTTTGCTTT
ACAACGTTGGTTAATTGCTGTTCGTAATGGTAAAACTTTAGCTGAAGCTATGCGTG-
GTTGGGTTCCTTTTGATGAATTATCTATTATTTCTGCTGGTGAAATTTCTGGTAAT-
GTTCATCAAGCTTTAGATGATATTATTTATATGAATGATACTAAAAAAAAGTTA
AA **\*** GGTTCTGGTTCTGGTTCTGGT **\*** ACTTTATCTCGTTGGACTGGTCGTTTAC-
GTTTATTTTTTGATCGTTTTATTCCTTGGTCTATTTATAAAACTATTATTGG
TTGTGGTTTTTTATTATCTTTAGCTTCTTTAATTAATGCTGGTATTCCTG
TTCCTGAAGCTTTACGTATTATTATGAAAACTGCTTCTCCTTGGTATAAAGAAC-
GTTTAGTTGCTATTCGTTATCGTTTATTAAATGGTGAACGTAATTTAG
GTGAAGCTTTAAATTCTTCTGGTTATATTTTTCCTTCTAAACAAATGATTATG-
GATTTACGTTCTTATGCTGCTTTAGGTGGTTTTTGATGAAATGTTAAATAAATTA
TCTGTTCAATGGCAAGATGATTCTGTTGCTTATATTACTAAACAAATGACTTA-
ACTCGAGCACCACCACCACCACCACTGAGATCCGGCTGCTAACAAAGCCCGAAAG-
GAAGCTGA **\*** ggt tct cca ggt tct.

The deletion region shown underlined (only for visuals) marked by (\*) and the insert sequence, small case (only for visuals), added after third (\*). See the FC_class2 report in S2 File.

Input sequence format for "only insertion" primer designing in case the user does not need to delete a sequence; the insertion point is marked by two stars (\*\*) and the insert sequence is provided as above. See below example.

ACTCTGTCAAGATGGACCGGAAGGTTGCGTCTCTTCTTTGACAGATTTA
TTCCCTGGTCGATATACAAAACCATTATAGGATGTGGTTTCTTGTTATCA**\*\***CTG-
GCATCGCTTATTAATGCAGGTATCCCTGTACCGGAAGCCTTACGAATAATAAT-
GAAAACGGCAAGT\*ccgtggtataaggaaagattagttgcta.

Input sequence format for "only insertion" primer designing. If the user wants to delete some sequences, the deletion is marked by a star (\*) as described in the deletion insertion example above, while no sequence required after third (\*) See below example.

ACTCTGTCAAGATGGACCGGAAGGTTGCGTCTCTTCTTTGACAGATTTATTC-
CCTGGTCGATATACAAAACCATTATAGGATGTGGTTTCTTGTTATCA\*CTGGCA
TCGCTTATTAATGCAGGTATCCC\*TGTACCGGAAGCCTTACGAATAATAAT-
GAAAACGGCAAGT\*.

Fig 2 provides a general guide and interface of the FastCloneAssist tool during the primer designing.

Users should know that FastCloning is an efficient and fast method for producing clones of varying lengths. Our primer design tool "FastCloneAssist" can generate primers for inserts of any length. However, since cloning relies on PCR, it is subject to the limitations of PCR reactions, such as the error rate, fidelity, and the maximum length of amplifiable DNA associated with the DNA polymerase used. For more information about the FastCloning method, please follow Li et al. (2011).

## Results and applications

The FastCloneAssist tool has been tested under various platforms such as Visual studio (Microsoft Corporation, Version 1.87.2), PyCharm (JETBrains), Jupytor Notebook and Google Colab (Google Colaboratory, [7]). We have tested more than 20 primer pairs designed using this tool (Table 1). We find that the FastCloneAssist tool has significant advantages over the manual method of primer design. FastCloneAssist generates primers almost instantaneously upon input, while traditional methods typically take a minimum of one hour and can be extended for several hours, depending on the regional complexity of the DNA sequence and

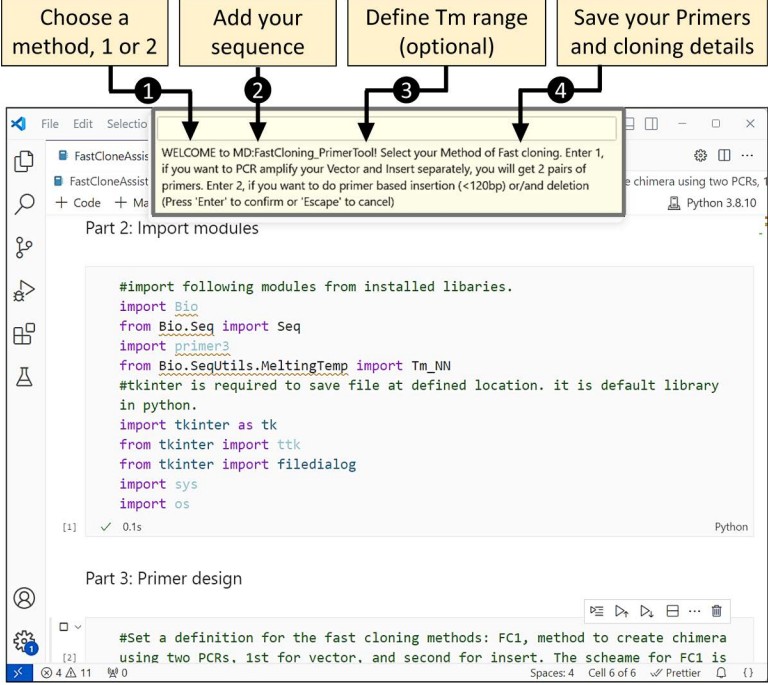

**Fig 2. Simple steps to use the FastCloneAssist tool.** To use the FastCloneAssist tool, start by running the program and selecting your preferred FastCloning method by entering 1 or 2. Next, input your DNA sequence in the specified format and optionally provide your desired melting temperature (Tm) value. Once you've entered the required information, the tool will process the inputs, and within a few seconds, a new window will appear, prompting you to save the results as a text file.

researchers' experience. Additionally, as the FastCloning method is primer-dependent, an error in primer design leads to corresponding errors in the construct, which may be detected after further delay for sequencing or other downstream processing, potentially adding weeks or months to the process. By ensuring error-free primer designing, FastCloneAssist provides significant time and cost saving advantages over the manual method of primer design.

The detailed user step-by-step protocol (S3 and S4 Files), script (S3 and S4 Files), example sequences (S7 File), and example result files are available in S1 and S2 Files and on the GitHub link..

GitHub link: https://github.com/ps-vcu/FastCloneAssist.git

## Case study

The FastCloning method was very useful to our ongoing mutagenesis project where we were creating various chimera genes and mutants to explore the biophysical properties of a set of proteins, BfpE, BfpD, BfpC, BfpB and BfpU from the enteropathogenic *Escherichia coli* type IV pilus, known as the bundle forming pilus (BFP) [12,13]. The method reduced the total time and cost of chimera/ mutant creation. In contrast, manual design and verification of Fast-Cloning primers required considerable additional effort, which led us to design the FastCloneAssist tool. The FastCloneAssist significantly improved our primer designing capability and primer quality. Here we provide examples to illustrate how we used FastCloning to modify BFP genes/proteins.

In the case of BfpE we had created a recombinant protein (cpBfpE) by deleting transmembrane regions in an attempt to express a soluble protein; However, protein expression was

**Table 1. List of primers designed using FastCloneAssist tool.**

| S. No. | Primer name | Sequence (5' ◊ 3') | Purpose |
|---|---|---|---|
| 1 | cpBfpE_L1f | ggttctccaggttctACTTTATCTCGTTG-GACTGGTCGTTTA | Del[a]/ins[b]; Primer pair to delete L0 and add L1 |
| 2 | cpBfpE_L1r | TagaacctggagaaccTTTA-ACTTTTTTTTTAGTATCAT-TCATATAAATAATATCATCTA-AAGC | |
| 3 | cpBfpE_L2f | tctccaggtccaagcggtACTTTATCTC-GTTGGACTGGTCGTTTA | Del/ins; Primer pair to delete L0 and add L2 |
| 4 | cpBfpE_L2r | cgcttggacctggagaaccTTTA-ACTTTTTTTTTAGTATCAT-TCATATAAATAATATCATCTA-AAGC | |
| 5 | cpBfpE_L3f | ggtggcggtggttctACTTTATCTCGTTG-GACTGGTCGTTTA | Del/ins; Primer pair to delete L0 and add L3 |
| 6 | cpBfpE_L3r | TagaaccaccgccaccTTTA-ACTTTTTTTTTAGTATCAT-TCATATAAATAATATCATCTA-AAGC | |
| 7 | cpBfpE_L4f | gttctgctgctggttctggtgaat-tcACTTTATCTCGTTGGACTGGTC-GTTTA | Del/ins; Primer pair to delete L0 and add L4 |
| 8 | cpBfpE_L4r | gaaccagcagcagaaccagcagaaccTTTA-ACTTTTTTTTTAGTATCAT-TCATATAAATAATATCATCTA-AAGC | |
| 9 | cpE_L0_del_187-211_F | GTTCTGGTACTATTATTGGTTGT-GGTTTTTTATTATCTTTAGCT | Pair to Delete T143 – K165) in cpBfpE |
| 10 | cpE_L0_del_187-211_R | ATAATAGTACCAGAACCAGAAC-CAGAACCTTTAAC | |
| 11 | cpE_C-His_F | AAATGACTCTCGAGCACCAC-CACCACCACCACTG | Del; Primer pair to delete the stop codon after C-term T272. |
| 12 | cpE_C-His_R | TGCTCGAGAGTCATTTGTTTAG-TAATATAAGCAACAGAATCA | |
| 13 | cpE_N-His_F | GATATACCATGAAAGAAAAAT-TAAATCGTTTAT-TATTTACTTCTAAAACTC | Del: Primer pair to delete the His tag from N-terminus. |
| 14 | cpE_N-His_R | TCTTTCATGGTATATCTCCTTCT-TAAAGTTAAACAAAAT-TATTTCTAGA | |
| 15 | PKS61F | gttgcccgggctgctgtTCTGATACGGA-CAGAAAAATAATCTCAGG | Primer pair to insert FLAsH tag at BfpD 416-420. |
| 16 | PKS61R | cagcagcccgggcaacaACTTGC-TATATATTCTGTTAATGTTATGC-TACATTG | |
| 17 | FC_BfpB_S92C_F | CTGAGATgtAGTACACCTCTGG-GATTTGATGAAGTA | Prime pair to replace S92 with C from BfpB protein. |
| 18 | FC_BfpB_S92C_R | GTGTACTacATCTCAGAATAAT-TCCATGTACACCTTCTAAT | |
| 19 | FC_BfpU_S105C_F | TTAATGTgTATAATGA-CAACTCCTAAAAAAGGATTAT-GTGC | Prime pair to replace S105 with C from BfpU protein. |
| 20 | FC_BfpU_S105C_R | TCATTATAcACATTAAGTATATC-GTGTTTTCATTTGTATCATATGA | |

*(Continued)*

**Table 1.** (Continued)

| S. No. | Primer name | Sequence (5' ◊ 3') | Purpose |
|---|---|---|---|
| | | **Vector and insert PCR and Chimera creation** | |
| 21 | PMD22_05-vR | CATATGgctctgaaaatacaggttttcGT-GATG | Primer pair to amplify vector pET15TEV to clone N-term BfpD. |
| 22 | PMD22_06-vF | TAAGGATCCTGCGAGCTCT-GTCGA | |
| 23 | PMD22_07-iF | ctgtattttcagagcCATATGGTGAA-CAAAACCGAAAAAACCAGCG | Primer pair to amplify and clone the N-term BfpD into pET15TEV at NdeI and BamH I site. |
| 24 | PMD22_08-iR | GAGCTCGCAGGATCCTTACGG-CAGCAGATCCTGACCATTATT | |

[a]Del = deletion, [b]Ins = insertion

insufficient for our purposes. To troubleshoot expression, we needed to clone various versions, altering the linker (L0) length and sequences, deleting some additional sequence (T143 – K165) that could represent additional transmembrane sequences, and changing the His tag location from N-term to C-term. For all these variants we used the FastCloneAssist tool to design the primers and perform the cloning efficiently. Fig 3A shows the cpBfpE sequence and translation. The linker region L0 has been changed with L1, L2, L3 and L4; see the linker sequences below and primers (1-14) in Table 1.

   L1 = 5' ggt tct cca ggt tct 3'; (GSPGS)
   L2 = 5' ggt tct cca ggt cca agc ggt 3'; (GSPGPGG)
   L3 = 5' ggt ggc ggt ggt tct 3'; (GGGGS)
   L4 = 5' ggt tct gct ggt tct gct gct ggt tct ggt gaa ttc 3'; (GSAGSAAGSGEF)
   # Note the parenthesis contains corresponding protein sequence.

   In a similar approach (deletion and insertion), we used FastCloneAssist to insert a FLAsH tag (CCPGCC) [14] in the BfpD protein at the specific location (416-420) see Fig 3B, (Table 1, primers 15-16).

   We also used FastCloning and FastCloneAssist to create a point mutation in BfpU and BfpB at S105C and S92C respectively, see Fig 3 (C & D), (Table 1, primer 17-20).

   To ligate N-term BfpD into pET15TEV vector and avoid an additional restriction (NdeI) site present in the middle of the gene (underlined in the sequence), we used the FastCloning method and primers (Table 1, primer 21-24) were designed using FastCloneAssist, see Fig 3E.

   In brief FastCloning and FastCloneAssist revolutionized our approach to chimera and mutant creation, accelerating experimentation and enhancing precision. This case study illustrates the high efficacy (all 24 designed primers successfully produced clones, see Table 1) of this tool in facilitating complex genetic manipulations, paving the way for advancements in protein engineering and biophysical studies.

## User guides

FastCloneAssist can be run in a local Python environment as well as using the cloud computing option in Google Colab.

### To run in a Python environment installed locally

To utilize the FastCloneAssist script, download it from the GitHub repository or from S5 File and execute it using Python. The code is structured into three parts. Begin by running the 1st

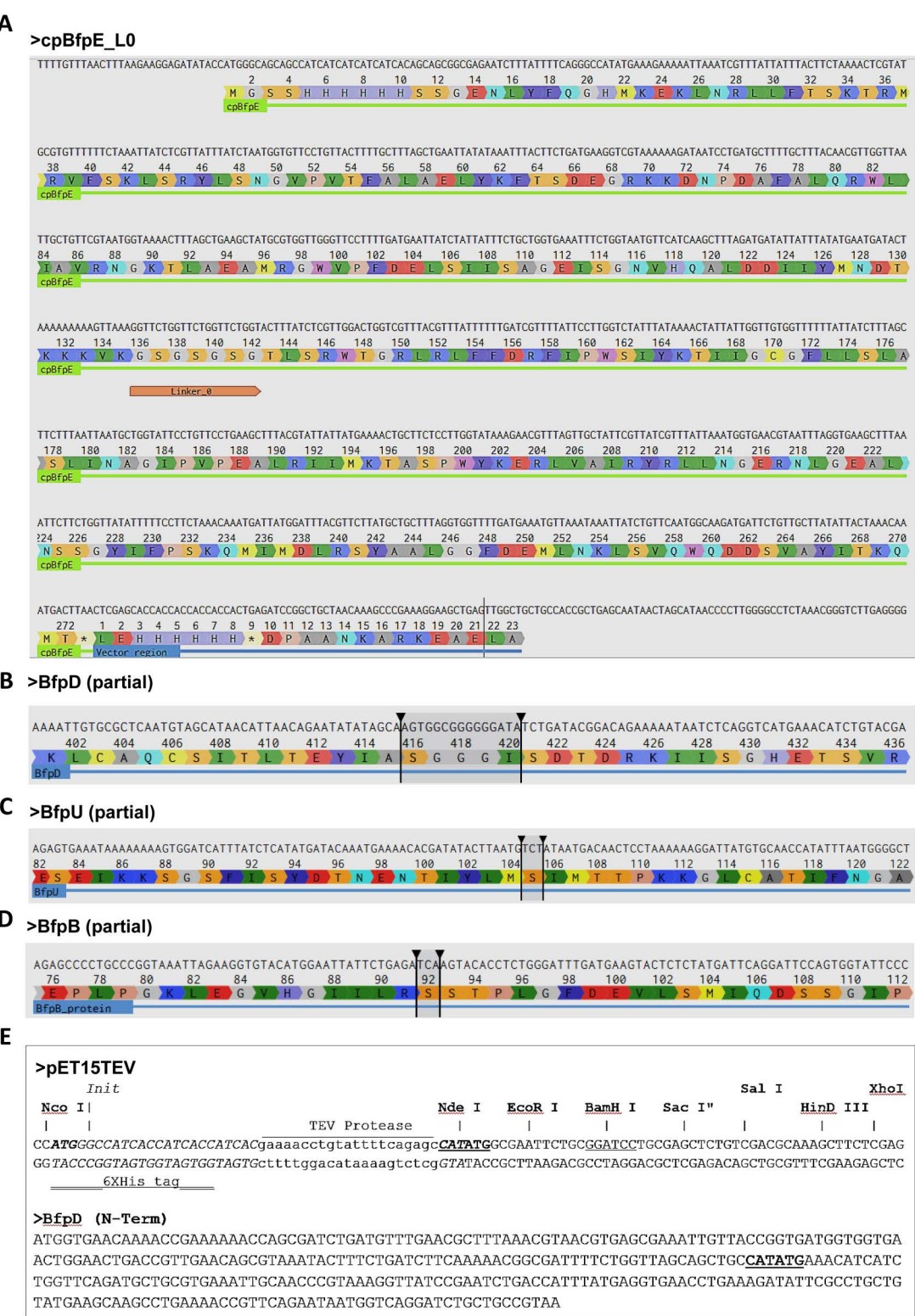

**Fig 3. Example sequences from the case study.** Showing DNA and protein sequence of BfpE, the linker L0 and His tag at N-terminus and C-terminus are depicted (A). (B, C, D), showing sequence of interest of BfpD, BfpU and BfpD respectively and the target deletion is selected by arrows. Cloning sites NdeI and BamHI (underlined) of pET15TEV and BfpD (N-Term) are shown in Fig (E).

part to install the required libraries (Biopython & primer3). It is important to note that the 1st part only needs to be executed once for a system, as it installs the necessary libraries. Once this step is completed, proceed to run the 2nd part. Part 2 will import all required modules from the installed libraries. Note the tkinter, sys, and os modules will be imported from the default Python installation.

Running the 3rd part (primer design part) the script will prompt you to enter 1 or 2, to choose either Class 1 or Class 2 style of primer designing (see above). Following your selection, the next required input will be the sequence. Ensure your sequence is pre-prepared with symbols as described for the specific class and need. The next prompts will ask if you want to choose any specific Tm or use the default range (55-65°C)? You may enter "no", but if your input is yes, it will prompt you to enter your Tm range, such as 60-64. After this input, the script will generate a report as a text file. See the S3 File for detailed user guide. Please minimize your open windows to see the popup asking you to save the file.

### To run on cloud using google Colab

The script is compatible with Google Colab as well. No Python installation is required in your system. To use it in Google Colab, create a Colab account using Google account. Download the Colab version of the FastCloneAssist script from the GitHub link or from S6 File and save it in your Google Drive. In Colab, import the script from your Google Drive and run it. You can right click on file "FastCloneAssist_Colab" file in your google drive and choose to run with Google Colaboratory. See the S4 File for detailed user guide for cloud version. Note, in Google Colab you need to run the 1st part each time you start a new session, as Google Cloud does not store the installed library once the session ends. The primer will be saved as a text file in your download folder.

## Conclusion

FastCloneAssist streamlines and expedites cloning primer design with its intuitive interface and robust algorithms, streamlining cloning experiments for researchers worldwide. By combining speed, accuracy, and customization options, it empowers users to achieve precise results efficiently. FastCloneAssist's user-friendly design and advanced features make it a go-to resource for genetic engineers, bioinformaticians, and molecular biologists. Harnessing its capabilities, researchers can expedite scientific breakthroughs and drive progress in the field. FastCloneAssist represents a significant advancement in primer design technology, poised to shape the future of molecular biology research. This tool holds the promise of further advancement of molecular biology and accelerating the pace of scientific discovery.

## Key Points

The innovation presented in this article is summarized as follows:

- **Streamlined FastCloning Primer Design:** Our manuscript introduces FastCloneAssist, a user-friendly Python program that automates primer design specifically for FastCloning. Researchers can forgo complex calculations; instead, simply providing their vector and insert sequences allows FastCloneAssist to handle the design process.

- **Precision PCR and Seamless Integration:** FastCloneAssist leverages established bioinformatic libraries to design primers that excel in two crucial areas. These primers ensure efficient PCR amplification while achieving seamless integration of the insert DNA into the desired vector position. This translates to fewer failed experiments and more successful cloning efforts.

- **Democratizing FastCloning:** FastCloneAssist is an open-source tool with minimal user expertise required. This broadens access to the powerful FastCloning technique, making it accessible to a wider range of researchers regardless of experience level. By streamlining workflows, FastCloneAssist accelerates scientific discovery across various disciplines.

## Supporting information

**S1 File. Example Report for FC_Class 1.txt.**
(TXT)

**S2 File. Example Report for FC_Class 2.txt.**
(TXT)

**S3 File. Step-by-step protocol to use the FastCloneAssist in local Python Environment. docx.**
(DOCX)

**S4 File. Step-by-step protocol to use the FastCloneAssist in Google Colab environment. docx.**
(DOCX)

**S5 File. Script_FastCloneAssist_VS02.ipynb.**
(IPYNB)

**S6 File. Script_FastCloneAssist_Colab_VS02.ipynb.**
(IPYNB)

**S7 File. Example sequence and input format.docx.**
(DOCX)

## Acknowledgments

The work described in this manuscript was funded by an internal "Accelerate Grant" from Virginia Commonwealth University (no award number).

## Author contributions

**Conceptualization:** Pradip Kumar Singh.

**Data curation:** Pradip Kumar Singh.

**Formal analysis:** Pradip Kumar Singh.

**Funding acquisition:** Michael S. Donnenberg.

**Investigation:** Pradip Kumar Singh, Michael S. Donnenberg.

**Methodology:** Pradip Kumar Singh.

**Resources:** Pradip Kumar Singh.

**Software:** Pradip Kumar Singh.

**Supervision:** Michael S. Donnenberg.

**Validation:** Pradip Kumar Singh.

**Visualization:** Pradip Kumar Singh.

**Writing – original draft:** Pradip Kumar Singh.

**Writing – review & editing:** Pradip Kumar Singh, Michael S. Donnenberg.

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
