## [Decision Letter · Decision Letter 0]

8 Dec 2024

PONE-D-24-21943Revolutionizing Molecular Cloning: Introducing FastCloneAssist, a Streamlined Python Tool for Optimizing Primer Design in Restriction & Ligation-Independent PCR CloningPLOS ONE

Dear Dr. Donnenberg,

Thank you for submitting your manuscript to PLOS ONE. After careful consideration, we feel that it has merit but does not fully meet PLOS ONE’s publication criteria as it currently stands. Therefore, we invite you to submit a revised version of the manuscript that addresses the points raised during the review process. Please submit your revised manuscript by Jan 22 2025 11:59PM. If you will need more time than this to complete your revisions, please reply to this message or contact the journal office at plosone@plos.org . Please include the following items when submitting your revised manuscript:

We look forward to receiving your revised manuscript.

Kind regards,

Mojtaba Kordrostami, Ph.D.

Academic Editor

PLOS ONE

Journal Requirements:

3. Please expand the acronym "VCU" (as indicated in your financial disclosure) so that it states the name of your funders in full.

Reviewers' comments:

Reviewer's Responses to Questions

**Comments to the Author**

1. Is the manuscript technically sound, and do the data support the conclusions?

Reviewer #1: Partly

2. Has the statistical analysis been performed appropriately and rigorously? 

Reviewer #1: I Don't Know

3. Have the authors made all data underlying the findings in their manuscript fully available?

Reviewer #1: Yes

4. Is the manuscript presented in an intelligible fashion and written in standard English?

Reviewer #1: Yes

5. Review Comments to the Author

Reviewer #1: The authors have meticulously performed the experimental procedure to design primers for PCR based Fast Cloning; but some issues have to be addressed before acceptance.

1. What is the utility of this web tool for designing primers for different length of inserts and different size of vectors. As the size is one of the key factors for cloning.

2. The manuscript would be more attractive if authors provide some data on success rate in cloning using these primers.

3. Is it possible to do the location specific insertion, deletion and mutation using these primers, without changing the reading frame.

4. I would also suggest to improve the picture quality.

Best of luck

6. PLOS authors have the option to publish the peer review history of their article (what does this mean? ). If published, this will include your full peer review and any attached files.

**Do you want your identity to be public for this peer review?** For information about this choice, including consent withdrawal, please see our Privacy Policy .

Reviewer #1: **Yes: ** ANIRUDHA CHATTOPADHYAY

---

## [Author Response · Author response to Decision Letter 1]

3 Jan 2025

All concerns were addressed as detailed in the response to reviewers file.

---

## [Decision Letter · Decision Letter 1]

7 Feb 2025

Revolutionizing Molecular Cloning: Introducing FastCloneAssist, a Streamlined Python Tool for Optimizing Primer Design in Restriction & Ligation-Independent PCR Cloning

PONE-D-24-21943R1

Dear Dr. Donnenberg,

We’re pleased to inform you that your manuscript has been judged scientifically suitable for publication and will be formally accepted for publication once it meets all outstanding technical requirements.

Kind regards,

Mojtaba Kordrostami, Ph.D.

Academic Editor

PLOS ONE

Additional Editor Comments (optional):

The manuscript can be accepted now

Reviewers' comments:

Reviewer's Responses to Questions

**Comments to the Author**

1. If the authors have adequately addressed your comments raised in a previous round of review and you feel that this manuscript is now acceptable for publication, you may indicate that here to bypass the “Comments to the Author” section, enter your conflict of interest statement in the “Confidential to Editor” section, and submit your "Accept" recommendation.

Reviewer #1: All comments have been addressed

2. Is the manuscript technically sound, and do the data support the conclusions?

Reviewer #1: Yes

3. Has the statistical analysis been performed appropriately and rigorously? 

Reviewer #1: Yes

4. Have the authors made all data underlying the findings in their manuscript fully available?

Reviewer #1: Yes

5. Is the manuscript presented in an intelligible fashion and written in standard English?

Reviewer #1: Yes

6. Review Comments to the Author

Reviewer #1: The manuscript entitled 'Revolutionizing Molecular Cloning: Introducing FastCloneAssist, a Streamlined Python

Tool for Optimizing Primer Design in Restriction & Ligation-Independent PCR Cloning' represents the FastClone assist tool for designing primer for PCR based cloning. Hopefully, the tool will be available freely to researchers. Good job

7. PLOS authors have the option to publish the peer review history of their article (what does this mean? ). If published, this will include your full peer review and any attached files.

**Do you want your identity to be public for this peer review?** For information about this choice, including consent withdrawal, please see our Privacy Policy .

Reviewer #1: **Yes: ** ANIRUDHA CHATTOPADHYAY

---

## [Editor Report · Acceptance letter]

PONE-D-24-21943R1

PLOS ONE

Dear Dr. Donnenberg,

I'm pleased to inform you that your manuscript has been deemed suitable for publication in PLOS ONE. Congratulations! Your manuscript is now being handed over to our production team.

Kind regards,

on behalf of

Dr. Mojtaba Kordrostami

Academic Editor

PLOS ONE